# Lightweight Model Adaptation for Mitigating Bias in Deep Learning Models for Chest X-Ray Analysis

**Clemence Mottez**[1,2] (iD)                                      CMOTTEZ@STANFORD.EDU

[1] *Center for Artificial Intelligence in Medicine and Imaging, Stanford University*

[2] *Institute of Computational and Mathematical Engineering, Stanford University*

**Louisa Fay**[1]                                                LFAY@STANFORD.EDU

**Jean-Benoit Delbrouck**[1]                                     JBDEL@STANFORD.EDU

**Curtis Langlotz**[1]                                           LANGLOTZ@STANFORD.EDU

**Editors:** Accepted for publication at MIDL 2025

## Abstract

Deep learning models have demonstrated significant potential in improving chest X-ray diagnosis. However, these models may exacerbate healthcare disparities. Addressing the inherent biases of deep learning models is essential to ensure their safe and reliable deployment in clinical practice. We suggest a novel bias mitigation approach that combines embeddings extracted by a Convolutional Neural Network with an eXtreme Gradient Boosting classifier. Our results show that this hybrid model significantly reduces bias across the sensitive attributes sex, age, and race, while maintaining comparable overall diagnostic performance and without the need for expensive model retraining. Our approach demonstrates that integrating simple, interpretable, and computationally efficient modifications into existing models can effectively enhance fairness in medical image analysis.

**Keywords:** Bias Mitigation, Chest X-ray, eXtreme Gradient Boosting

## 1. Introduction

Deep Learning (DL) models have the potential to transform healthcare by increasing diagnostic accuracy, personalizing treatment, and improving patient outcomes(Alowais et al., 2023). However, these technologies risk exacerbating healthcare disparities if their performance varies across different subgroups of patients, for example according to sex, age, and race (Yang Y., 2024). These biases may arise from training data that underrepresents certain populations, algorithm designs that overlook the unique characteristics of different groups, or disparities in healthcare access(Gianfrancesco, 2018). Biases are among the many barriers that prevent the deployment of these models in clinical practice, where equitable outcomes are crucial(Wiens, 2019). Current bias mitigation methods involve tradeoffs between fairness and accuracy. Techniques such as rebalancing training datasets or modifying algorithms often require extensive model retraining(Yang et al., 2024) and are thus impractical in healthcare due to data scarcity and resource constraints. In response, our study proposes a novel lightweight model adaptation strategy to mitigate biases related to sex, age, and race in Chest X-ray (CXR) diagnosis. Our solution is to replace the final classification layer of a Convolutional Neural Network (CNN) with an eXtreme Gradient Boosting (XGBoost) model which is then retrained on a curated subset of data. This hybrid approach leverages CNNs' feature extraction capabilities and XGBoost's effectiveness

in handling class imbalances to reduce bias. While previous studies have shown better classification performance when the last layer of CNNs is replaced by an XGBoost classifier(Shanmugam, 2023; Sugiharti et al., 2022; Hedhoud et al., 2023), none have explored this combination to mitigate bias. Our hybrid CNN-XGBoost framework demonstrates that equitable diagnostics are achievable without sacrificing accuracy, reducing bias across sex, age, and race subgroups by 62.6% while improving the overall performance by 8.9%.

## 2. Method

We use the DenseNet-121 model from TorchXrayVision(Cohen et al., 2021) pretrained on the CheXpert(Irvin et al., 2019) dataset to encode each X-ray image. For each image, we first extract features from the last hidden layer of the CNN, resulting in a 1024-dimensional embedding. Next, we reduce their dimension with reduction techniques including Principal Component Analysis (PCA) and encoder-decoder architectures. We select the method that maximizes performance and minimizes bias on the validation data. The resulting vector serves as input to an XGBoost classifier, which is then trained. XGBoost was chosen due to its ensemble nature. Multiple trees are trained to correct the errors of the previous trees, inherently focusing on harder-to-classify examples such as underrepresented groups. We expect this integration to both reduce the performance gap across subgroups and increase the overall performance. The pipeline is presented in Figure 1A.

This method is model-agnostic, which means that it can be adapted to other model architectures designed for image feature extraction. Moreover, existing bias mitigation techniques, including adversarial training, sample weighting, and data augmentation(Yang et al., 2024) can easily be integrated into our framework by only retraining the XGBoost head.

## 3. Experiments and Results

**Datasets:** We evaluate our method on two datasets to ensure the robustness and generalization of our model across different clinical environments. First, in-distribution data from CheXpert(Chambon, 2024), a dataset consisting of 224,316 CXRs obtained at Stanford Health Care. Second, Out-Of-Distribution (OOD) data from Medical Information Mart for Intensive Care (MIMIC)(Johnson, 2019) comprising 377,110 CXRs performed at the Beth Israel Deaconess Medical Center. Detailed dataset information are presented in Figure 1B.
**Embedding Analysis:** We first analyzed the embeddings extracted with DenseNet using PCA, t-distributed Stochastic Neighbor Embedding (t-SNE)(Van der Maaten, 2008) plots, and statistical tests such as the two-sample Kolmogorov–Smirnov test. Results, as shown in the Appendix Figure 2, indicate significant differences across subgroups, suggesting that the model could use shortcuts for disease classification, potentially leading to biased results.
**Embeddings reduction:** Results in the Appendix Table 1 show that reducing the size of the embedding using PCA to select the components that retain 95% of the total variability leads to a larger decrease in bias while maintaining a competitive overall performance.
**Bias Mitigation:** We focused the analysis on pleural effusion, due to its clinical significance and prevalence in the datasets. We used the Area Under Precision-Recall Curve (AUPRC) as the primary performance metric, due to its effectiveness in imbalanced data and in balancing precision and recall. We assessed the presence of bias related to the subgroups using $\Delta$AUPRC. For sex, we focused on the difference in performance between

males and females; for age, we used a threshold of 70 years old; for race, we focused our analysis on White, Black, and Asian. Each experiment is run five times and results are averaged. The XGBoost training parameters are described in the Appendix 4. As shown in Figure 1C, the original CNN model leads to higher performance differences between the subgroups than our novel approach. By incorporating XGBoost, the model leads to fairer and more consistent results. Results show a decrease in bias of 79.2% for sex, 47.1% for age, and 61.7% for race while improving the overall performance by 8.9%.

**Other classification heads:** Linear regression, decision trees, random forests, and balanced random forests were also tested. However, they either resulted in worse overall performance or in a smaller reduction in bias in comparison to an XGBoost head.

**OOD adaptability:** We evaluate our method by applying a DenseNet model pretrained on the CheXpert dataset to the OOD MIMIC dataset, assessing both performance and bias mitigation consistency. While the classification performance on the MIMIC dataset is lower than on the CheXpert dataset, our results demonstrate that our approach generalizes well OOD, reducing overall bias by 39.2% and improving overall performance by 8.2%.

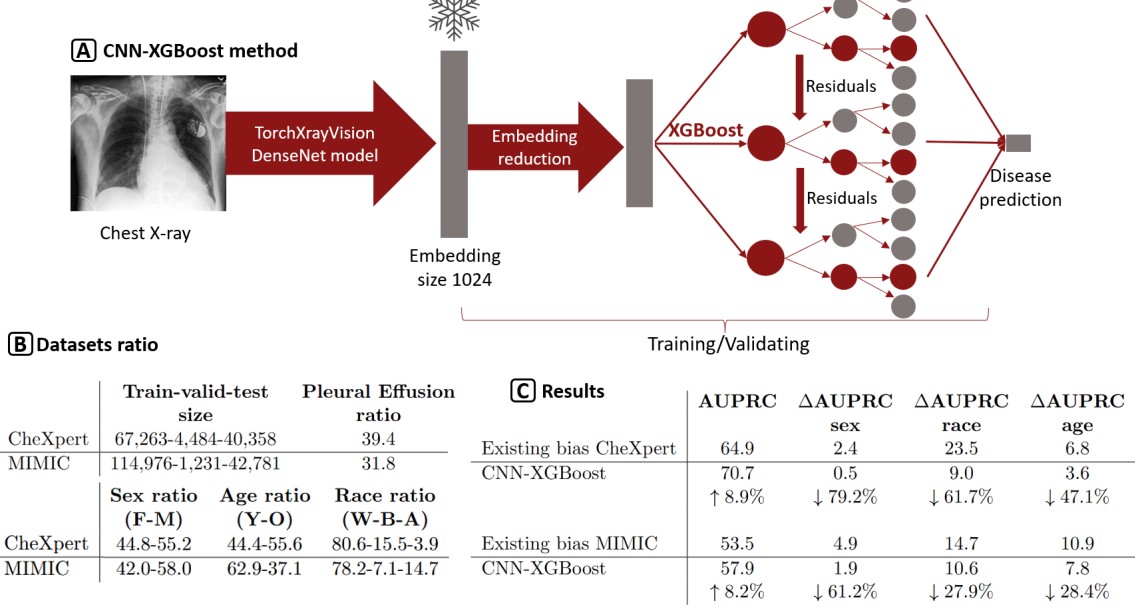

**B Datasets ratio**

|  | Train-valid-test size | Pleural Effusion ratio |
|---|---|---|
| CheXpert | 67,263-4,484-40,358 | 39.4 |
| MIMIC | 114,976-1,231-42,781 | 31.8 |

|  | Sex ratio (F-M) | Age ratio (Y-O) | Race ratio (W-B-A) |
|---|---|---|---|
| CheXpert | 44.8-55.2 | 44.4-55.6 | 80.6-15.5-3.9 |
| MIMIC | 42.0-58.0 | 62.9-37.1 | 78.2-7.1-14.7 |

**C Results**

|  | AUPRC | ΔAUPRC sex | ΔAUPRC race | ΔAUPRC age |
|---|---|---|---|---|
| Existing bias CheXpert | 64.9 | 2.4 | 23.5 | 6.8 |
| CNN-XGBoost | 70.7 | 0.5 | 9.0 | 3.6 |
|  | ↑8.9% | ↓79.2% | ↓61.7% | ↓47.1% |
| Existing bias MIMIC | 53.5 | 4.9 | 14.7 | 10.9 |
| CNN-XGBoost | 57.9 | 1.9 | 10.6 | 7.8 |
|  | ↑8.2% | ↓61.2% | ↓27.9% | ↓28.4% |

Figure 1: (A) Our CNN-XGBoost pipeline, (B) datasets information, (C) results (AUPRC: higher is better ; ΔAUPRC: lower is better).

## 4. Conclusion and Future Work

Our hybrid bias-reduction method improves performance and mitigates bias related to sex, age, and race in pleural effusion prediction from CXR images. By using a model-agnostic approach, the integration can be applied to existing CNN models without the need for retraining, which is beneficial for already trained models. Future work includes extending this analysis to other model architectures, such as transformers, analyzing additional chest medical conditions, and combining other bias mitigation strategies with our method.

## Acknowledgment

This work was supported in part by the Medical Imaging and Data Resource Center (MIDRC), which is funded by the National Institute of Biomedical Imaging and Bioengineering (NIBIB) under contract 75N92020C00021 and through the Advanced Research Projects Agency for Health (ARPA-H).

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

## Appendix

Table 1: Classification performance and bias on validation set when reducing the image embeddings using an Encoder-Decoder architecture, a PCA framework, or keeping the original embedding size.

| Encoder-Decoder Output dimension | AUPRC | ΔAUPRC sex | ΔAUPRC race | ΔAUPRC age |
|---|---|---|---|---|
| 64 | 70.8 | 3.2 | 19.8 | 6.8 |
| 128 | 71.4 | 2.2 | 14.9 | 6.4 |
| 256 | 71.7 | 2.6 | 17.9 | 5.4 |
| **PCA** Variance retained (in %) | AUPRC | ΔAUPRC sex | ΔAUPRC race | ΔAUPRC age |
| 98 | 70.8 | 1.7 | 16.6 | 4.8 |
| 95 | 71.1 | 1.2 | **14** | 5.1 |
| 90 | 71.9 | **0.7** | 18 | 5.5 |
| 85 | 71.1 | 3.2 | 14.7 | **4.2** |
| **Original size** | AUPRC | ΔAUPRC sex | ΔAUPRC race | ΔAUPRC age |
| | **72.7** | 2 | 17.7 | 5.9 |

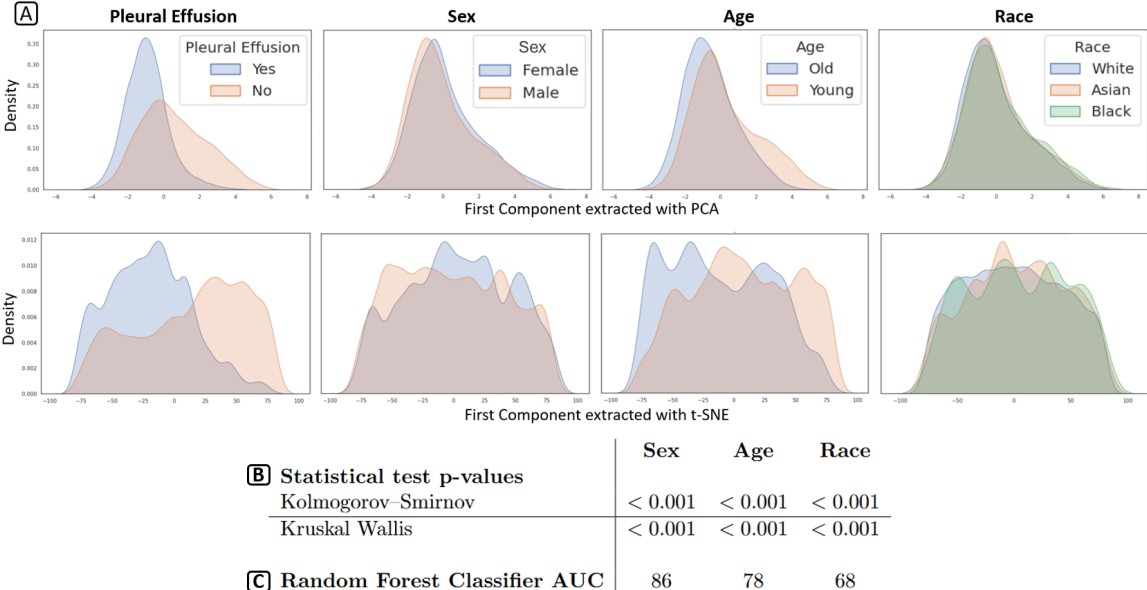

Figure 2: (A) Densities of the embeddings reduced using PCA (top row) and t-SNE (bottom row) according to the different values of pleural effusion and the sensitive attributes, (B) statistical significance (p-values) of the difference between subgroups for all sensitive attributes obtained from different statistical tests, (C) Area Under the Curve (AUC) of a Random Forest Classifier evaluating the ability of embeddings to predict the sensitive attributes.

**XGBoost Hyperparameters:** The hyperparameters were selected based on model performance evaluated on the validation dataset. The optimized hyperparameters are as follows: `eval_metric = 'logloss'`, `learning_rate = 0.05`, `n_estimators = 150`, and `max_depth = 10`.

