# OpenReview forum: "Lightweight Model Adaptation for Mitigating Bias in Deep Learning Models for Chest X-Ray Analysis"
_MIDL.io/2025/Short_Papers — MIDL 2025 - Short Papers_

### Official Review · Reviewer_6h7H · 2025-04-26

**Rating:** 4
**Confidence:** 4

**Summary:**

This paper proposes a novel approach to mitigate biases in deep learning models for chest X-ray analysis without requiring expensive model retraining. The authors introduce a hybrid model that combines the feature extraction capabilities of CNNs with an XGBoost classifier replacing the final classification layer. Their experiments focus on pleural effusion detection across two datasets, evaluating bias reduction across sex, age, and race dimensions. The results demonstrate that this approach reduces performance disparities between demographic subgroups by 62.6% while simultaneously improving overall accuracy by 8.9%. The method is model-agnostic and can be applied to existing pre-trained models, making it a practical solution for clinical applications where resources for complete model retraining are limited.

**Strengths:**

- The approach is practical and computationally efficient, providing a simple yet effective way to reduce bias without complete model retraining.
- The paper demonstrates solid empirical results showing significant bias reduction (79.2% for sex, 47.1% for age, 61.7% for race) while simultaneously improving overall performance.
- The methodology was validated on both in-distribution and out-of-distribution data, showing good generalization capabilities.

**Weaknesses:**

- The paper lacks detailed explanation of why XGBoost specifically helps mitigate bias compared to other potential classifiers.
- There is limited discussion on potential trade-offs between fairness and other metrics beyond AUPRC (such as specificity or sensitivity).

---

### Decision · Program_Chairs · 2025-05-01

Accept